

# Drifting Dynamics of the Bluebottle

Daniel Lee[1,2], Amandine Schaeffer[1,2], and Sjoerd Groeskamp[3]

[1]Coastal and Regional Oceanography Lab, School of Mathematics and Statistics, UNSW Australia, Sydney, New South Wales, Australia
[2]Centre for Marine Science and Innovation, UNSW Australia, Sydney, New South Wales, Australia
[3]NIOZ Royal Netherlands Institute for Sea Research, Texel, Netherlands

**Correspondence:** Daniel Lee (d.i.lee@unsw.edu.au)

**Abstract.** *Physalia physalis*, also called the Bluebottle in Australia, is a colonial animal resembling a jellyfish that is well known to beachgoers for the painful stings delivered by their tentacles. Despite being a common occurrence, the origin of the Bluebottle before reaching the coastline is not well understood, and neither is the way it drifts at the surface of the ocean. Previous studies used numerical models in combination with simple assumptions to calculate the drift of this species, excluding complex drifting dynamics. In this study, we provide a new parametrization for Lagrangian modelling of the Bluebottle by considering the similarities between the Bluebottle and a sailboat. This allows us to compute the hydrodynamic and aerodynamic forces acting on the Bluebottle and use an equilibrium condition to create a generalised model for calculating the drifting speed and course of the Bluebottle under any wind and ocean current conditions. The generalised model shows that the velocity of the Bluebottle is a linear combination of the ocean current velocity and the wind velocity scaled by a coefficient ('shape parameter') and multiplied by a rotation matrix. Adding assumptions to this generalised model allows us to retrieve models used in previous literature. We discuss the sensitivity of the model to different parameters (shape, angle of attack and sail camber) and explore different cases of wind and current conditions to provide new insights into the drifting dynamics of the Bluebottle.

## 1 Introduction

*Physalia physalis* (Fig. 1), also called the Indo-Pacific Portuguese man-of-war or the Bluebottle (*Physalia utriculus*, a synonym), is well known on the east coast of Australia for stinging tens of thousands of beachgoers each year (Surf Life Saving Australia: Sydney (2020)). The species is found throughout the world's oceans, in tropical, subtropical and (occasionally) temperate regions (Munro et al. (2019)). The Bluebottle resembles a jellyfish but is actually a siphonophore, a colonial organism composed of small individual animals called zooids (Totton and Mackie (1960)). There are four zooids depending on each other for survival and performing different functions, such as digestion (gastrozooids), reproduction (gonozooids) and hunting (dactylozooids). The last zooid, the pnumatophore, is a gas-filled float or sac that supports the other zooids and acts like a sail so the Bluebottle is constrained to the ocean surface, moving at the mercy of the wind, waves and marine currents. The Bluebottle's long tentacles hang below the float as they drift, fishing for prey to sting and drag up to their digestive zooids (Totton and Mackie (1960)).


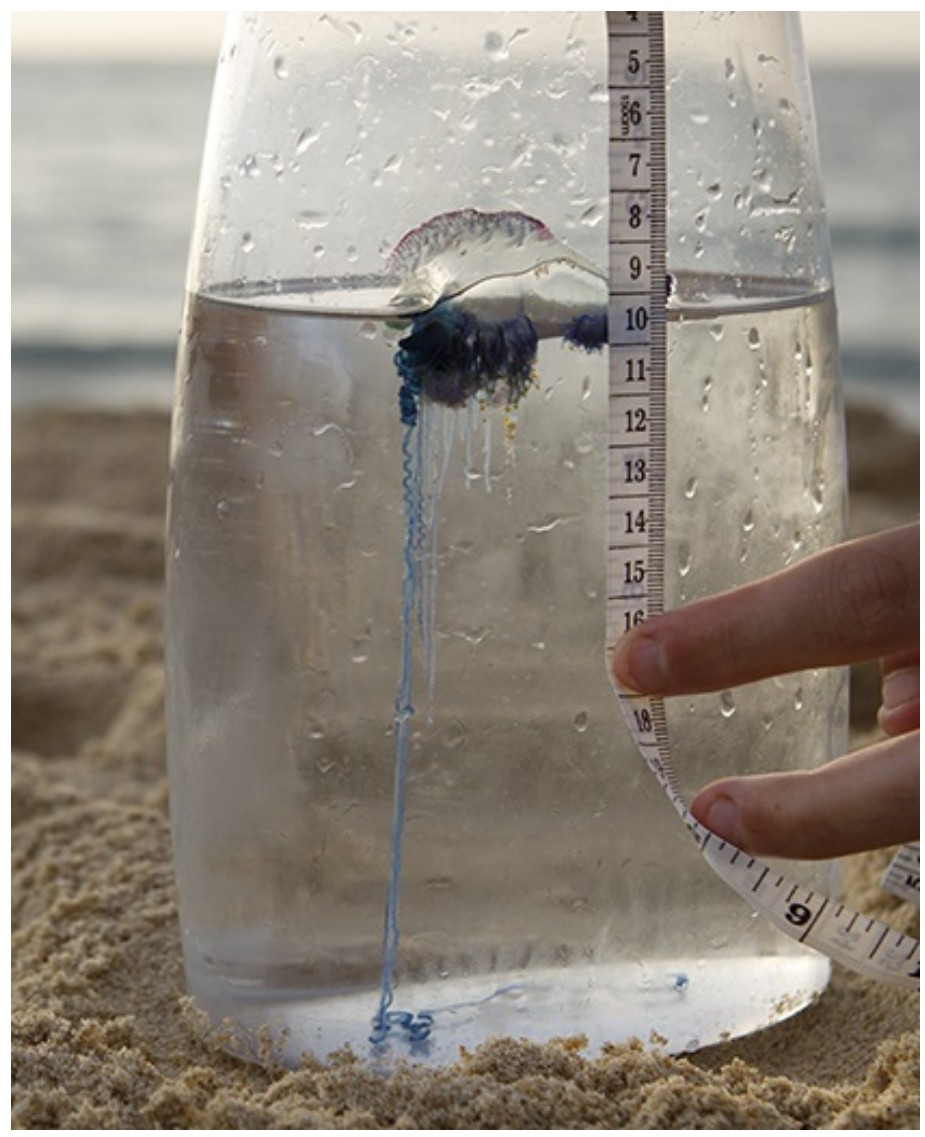

**Figure 1.** Photograph of a Bluebottle taken at Coogee Beach, Sydney. Courtesy of Lisa Clarke.

For each Bluebottle the float can be oriented towards the left or the right (dimorphism), believed to be an adaptation that prevents the entire population from being washed on shore to die (Totton and Mackie (1956), Woodcock (1944)). The "left-handed" Bluebottles sail to the right of the wind, while the "right-handed" Bluebottles sail to the left. The wind will always push the two types of Bluebottles in different directions, so at most half the population will be pushed towards the coast (Totton and Mackie (1956), Woodcock (1944)). The Atlantic Portuguese man-of-war (PMW) is considered the same species as

the Bluebottle, but with key differences in their size and the number of long tentacles used for hunting. The Bluebottle's float





rarely exceeds 10 cm and it has one long hunting tentacle that is less than 3 m in length (Fig. 1). In comparison, the PMW has floats of 15 cm and several hunting tentacles that can reach 10 m in length.

Due to their inability to swim, the movement of the Bluebottle can be modelled by calculating the forces acting on it, or by advecting virtual particles in ocean and atmospheric circulation models. Previous studies modeled the movement of the PMW
with Lagrangian particle tracking to explain major beaching events. For example, Ferrer and Pastor-Rollan (2017) where able to estimate the region of origin of a significant beaching event on the Basque coast in August 2010. They ran a Lagrangian model backwards in time, using wind velocity ($\mathbf{V}_A$) and a wind drag coefficient ($\lambda = 4.5\%$) as drivers of the PMW motion ($\mathbf{V}_{bb} = \lambda\mathbf{V}_A$). They found that the region of origin was the North Atlantic Subtropical Gyre. Prieto et al. (2015) included both the effect of the surface currents ($\mathbf{u}$) and wind ($\mathbf{V}_{bb} = \mathbf{u} + \lambda\mathbf{V}_A$) to predict initial colony position of major beaching events in
the Mediterranean in 2010. This model assumed the PMW was advected by the surface currents, with the effect of the wind being added with a much higher wind drag coefficient of 10%. Similarly, Headlam et al. (2020) used beaching and offshore observations to run a similar hindcast, using the joint effects of surface currents and wind drag, for the largest mass PMW beaching on the Irish coastline in over 150 years.

These previous models made the key assumption that the PMW's sailing direction is the same as the wind direction. This
may be based on the observation by Totton and Mackie (1960) that in winds stronger than force 4 (i.e. over 8m/s), the PMW would sail straight downwind with its sail parallel to the wind direction. However, it should be noted that this was a second-hand visual observation of one instance by a single individual. In addition, Totton and Mackie (1960) performed their own experiments and observed that in light winds the PMW balances itself at approximately $40^o$ to the wind (angle of attack), resulting in a completely different course relative to the wind. More recently, Ferrer and González (2020) improved on the
model from Ferrer and Pastor-Rollan (2017) by analysing the same beaching event but incorporating dimorphism and different drift angles relative to the wind direction. They found different regions of origin depending on the drifting angle considered, and concluded that the PMWs were likely right-handed.

Iosilevskii and Weihs (2009) took a different approach by expressing the forces acting on the PMW. By considering the equilibrium condition of the aerodynamic (above water) and hydrodynamic (below water) forces, when the velocity of the
PMW is constant, they derive equations that can be solved for the PMW's speed and direction of motion relative to the wind. However, they consider a situation with no background ocean current, where the hydrodynamic force is only the drag, opposed to the PMW course. Here we expand that model by adding the effect of the ocean current and simplifying the model down to an intuitive generalised vector form which could be implemented in Lagrangian models. We also analyze the impact of key variables such as angle of attack and sail camber (Sect. 4.4).

This paper is structured as follows. First we explain the methods and key assumptions used for our theoretical model (Sect. 2), followed by the force balance acting on the Bluebottle (Sect. 3). We then solve the equilibrium condition (Sect. 4.1), discuss the parameters in the model (Sect. 4.2 and 4.4) and apply the model to a few special cases that were chosen as instructive examples of the Bluebottle's sailing dynamics (Sect. 5). Finally, we compare our results to previous studies and discuss some variables that were not included in the model (Sect. 6).





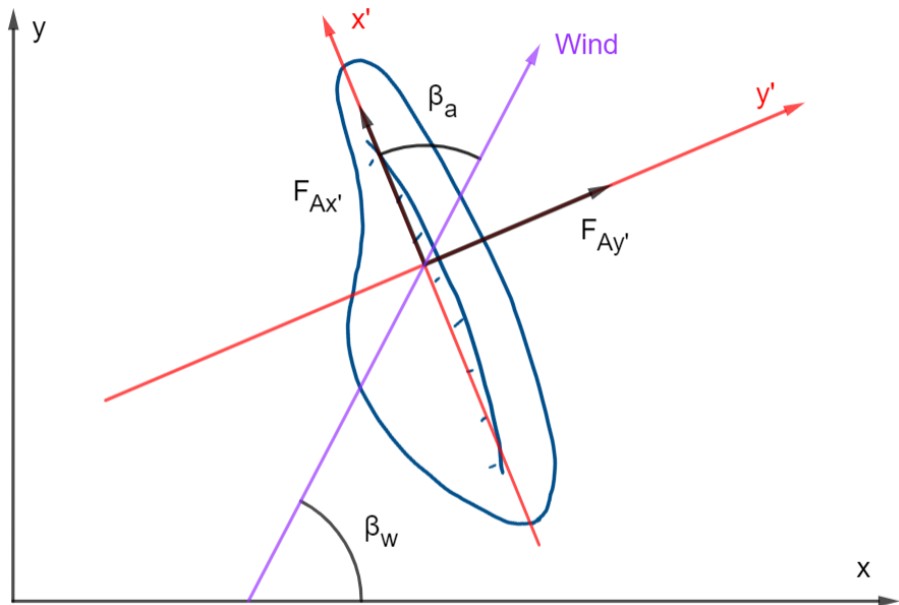

**Figure 2.** Top-down view of a left-handed (right-sailing) Bluebottle. x′, y′, x and y axes are defined above. $F_{Ax'}$ and $F_{Ay'}$ are the components of the aerodynamic force on the x′ and y′ axes respectively. $\beta_a$ is the angle of attack. $\beta_w$ is the angle of the wind.

## 2   Methods

The Bluebottle undergoes similar forces as a sailboat. Therefore, like Iosilevskii and Weihs (2009), we consider the aerodynamic and hydrodynamic forces acting on the Bluebottle, as with a sailboat (Szelangiewicz and Żelazny, 2018). The aerodynamic force, generated by the wind blowing against the Bluebottle's sail (Fig. 2), is split into two components, $F_{Ay'}$ (perpendicular to sail) and $F_{Ax'}$ (parallel to sail). The magnitude of the aerodynamic force is dependent on the wind speed, the area of the sail and the orientation of the Bluebottle to the wind, which we call the angle of attack $\beta_a$ (Fig. 2). We also refer to the angle between the x axis and the wind as $\beta_w$, using the standard x and y axes, which are east/west and north/south respectively.

The hydrodynamic force is caused by the interactions of the water with the submerged body of the Bluebottle. The wind-driven motion of the Bluebottle through the water creates a hydrodynamic drag opposite to the direction of motion. The sum of the wind-driven drag and the bluebottle motion due to the background ocean current results in a relative current, which determines the total hydrodynamic force acting on the Bluebottle. Compared to Iosilevskii and Weihs (2009), we here add the effect of the ocean current but we do not consider the drag caused by the tentacles (see Sect. 6) and assume that the submerged part of the Bluebottle is a cylinder. This is a reasonable assumption since the Bluebottle has only one tentacle which is much shorter than the PMW's tentacles. The submerged body is important for drag, but does not have a directional component to it because it is cylindrical. This differs from the submerged part of a sailboat, which has a long straight keel sticking down from the bottom of the boat. The keel minimizes sideways motion and rotation, allowing the boat to maximize forward motion. Since the Bluebottle's body has no keel and is assumed to be perfectly symmetrical, it does not restrict rotation and the orientation





of the Bluebottle will be completely determined by the wind. Furthermore, the Bluebottle's motion will not be restricted by a keel, so it can move forward, sideways, or at any other angle with respect to its sail.

The forces acting on the Bluebottle will be expressed as components on two axes where the force coefficients and affected
surface areas are most easily calculated. We call these the x′ and y′ axes and they are defined relative to the Bluebottle's sail. The x′ axis is along the chord of the sail. The y′ axis is perpendicular to the x′ axis and goes through the centre of the sail, which is also assumed to be symmetric (Fig. 2). We use the axes labels as subscripts for variables that are related to specific axes. Any vectors in this paper are labelled in bold text and their components are the standard x- and y- components. Angles that are measured anticlockwise are considered positive, while angles that are measured clockwise are considered negative.
The only exception to this is $\beta$, which is always considered positive (Fig. 3).

## 3  Forces acting on the Bluebottle

We now present the formulas that represent the aerodynamic and hydrodynamic forces acting on the Bluebottle. Compared to Iosilevskii and Weihs (2009), we choose not to consider the moment-force or the distances between the aerodynamic centre of effort and the hydrodynamic centre of effort, as these variables would require very rough estimates. Furthermore, Iosilevskii
and Weihs (2009) do not use these variables in the final equations that describe the course and speed, but rather to analyze the PMW's sail contraction and tilting of their tentacles - which we don't consider here. The moment-force would result in a torque, and thus the orientation of the Bluebottle would be influenced by a combination of the wind and current conditions. However, similarly to the leeway methodology (discussed further in Sect. 6), the orientation behaviour of the Bluebottle (mainly represented by the angle of attack) can instead be determined by observations made in physical experiments.

### 3.1  Aerodynamic force

The aerodynamic force on the Bluebottle $F_A$ is expressed as components on the x′ and y′ axes. This can be represented by the standard aerodynamic force equation, often used for lift and drag force on an aeroplane wing for instance.

$$\begin{cases} F_{Ax'} &= \frac{1}{2}\rho_A S_{x'} V_A^2 C_{Ax'} \\ F_{Ay'} &= \frac{1}{2}\rho_A S_{y'} V_A^2 C_{Ay'} \end{cases} \tag{1}$$

where $\rho_A$ is the density of the air (taken as $1.225$ kg m$^{-3}$), $S_{x'}$ and $S_{y'}$ are the areas of the sail in the respective axis, $C_{Ax'}$
and $C_{Ay'}$ are the respective force coefficients, and $V_A$ is the wind speed. Wind speed is used rather than relative wind speed because the speed of the Bluebottle is at least one order of magnitude smaller compared to the wind speed.

The areas $S_{x'}$ and $S_{y'}$ are fixed values for a particular Bluebottle. On the other hand, the force coefficients $C_{Ax'}$ and $C_{Ay'}$ are functions of the angle of attack $\beta_a$. Note that here we separately calculate the components of the aerodynamic force on two different axes. This is because the values of the x′ and y′ force coefficients and areas will vary significantly.

Considering the order of magnitude of the parameters, for wind speeds from 1 to 10 ms$^{-1}$, $F_{Ax'}$ will range from $7 \times 10^{-6}$ N to $7 \times 10^{-4}$ N and $F_{Ay'}$ will range from $1 \times 10^{-4}$ N to $1 \times 10^{-2}$ N.





## 3.2 Hydrodynamic force

The hydrodynamic force on the Bluebottle $F_H$ is dependent on the relative current, which is the current felt by the Bluebottle as it is in motion. The magnitude of the hydrodynamic force can be represented by an equation of the same form as the aerodynamic force.

$$F_{\mathrm{H}} = \frac{1}{2}\rho_{\mathrm{H}} S_{\mathrm{H}} V_{\mathrm{RH}}^2 C_{\mathrm{H}} \tag{2}$$

where $\rho_{\mathrm{H}}$ is the density of the water (taken as $1025\ \mathrm{kg\ m^{-3}}$), $S_{\mathrm{H}}$ is the projected area of the submerged Bluebottle surface onto the Bluebottle's plane of symmetry, $C_{\mathrm{H}}$ is the force coefficient, and $V_{\mathrm{RH}}$ is the speed of the current relative to the Bluebottle (Fig. 3). Considering the order of magnitude of the parameters, $F_{\mathrm{H}}$ will range from $1 \times 10^{-4}$ N to $1 \times 10^{-2}$ N.

The hydrodynamic force $F_{\mathrm{H}}$ is calculated using a single equation, thus using just one value for both the area and the force coefficient. Unlike the aerodynamic force, we do not need to calculate two components of the force separately. This is because the submerged body of the Bluebottle is close to cylindrical, so there is not much variance in the value of the area or force coefficient. The relative speed of the current can be represented by

$$V_{\mathrm{RH}} = \sqrt{V_{\mathrm{Hx}}^2 + V_{\mathrm{Hy}}^2} \tag{3}$$

$$\begin{cases} V_{\mathrm{RHx}} & = u - V_{\mathrm{bbx}} \\ V_{\mathrm{RHy}} & = v - V_{\mathrm{bby}} \end{cases} \tag{4}$$

where $V_{\mathrm{RHx}}$ and $V_{\mathrm{RHy}}$ are the components of the relative velocity of the current in the respective axes, $u$ and $v$ are the x and y components of the ocean current velocity, $V_{\mathrm{bbx}}$ and $V_{\mathrm{bby}}$ are the x and y components of the Bluebottle velocity vector.

To later solve for the equilibrium condition (Sect. 4.1), when the velocity of the Bluebottle is constant, we require the hydrodynamic force on the x′ and y′ axes. This is represented by

$$\begin{cases} F_{\mathrm{Hx'}} & = F_{\mathrm{H}} \cos\beta \\ F_{\mathrm{Hy'}} & = F_{\mathrm{H}} \sin\beta \end{cases} \tag{5}$$

where $\beta$ is the angle between the x′ axis and the relative velocity of the current (Fig. 3).




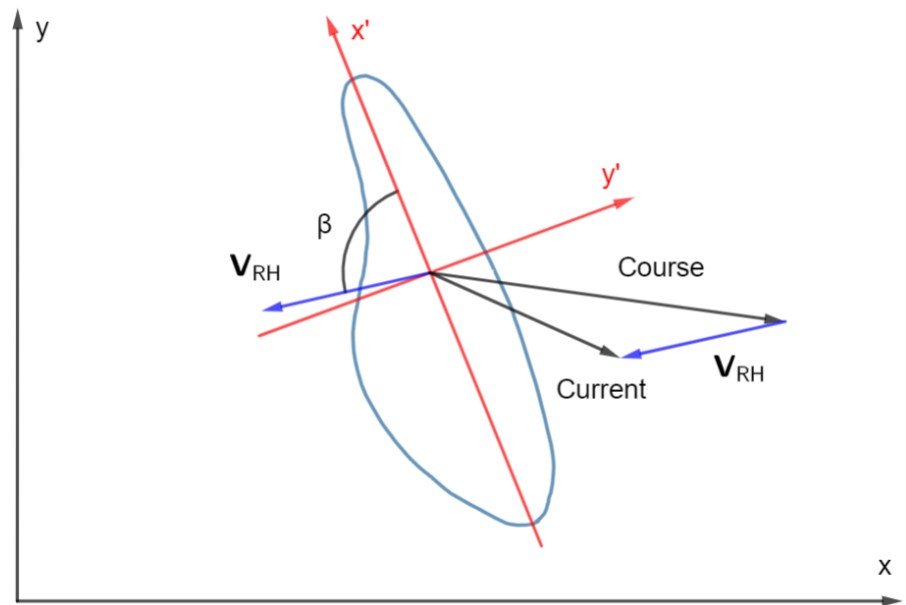

**Figure 3.** Relative current velocity $\mathbf{V}_{\mathrm{RH}}$ is the difference between the ocean current velocity and the Bluebottle's velocity (called Course). $\beta$ is the angle between the x$'$ axis and the relative velocity of the current.

## 4 Solving Bluebottle Velocity

### 4.1 Equations

We now solve for the equilibrium condition, where the velocity of the Bluebottle will be constant, hence there is no acceleration since the Bluebottle does not swim. Based on Newton's Second Law, this occurs when the net forces acting on the Bluebottle are zero. Hence, the aerodynamic and hydrodynamic forces must be equal in magnitude on each axis. Expressing the equilibrium conditions

$$\begin{cases} F_{\mathrm{Ax}'} & = F_{\mathrm{Hx}'} \\ F_{\mathrm{Ay}'} & = F_{\mathrm{Hy}'} \end{cases} \tag{6}$$

along the x$'$ and y$'$ axes yields the equations

$$\begin{cases} \rho_{\mathrm{A}} S_{\mathrm{x}'} V_{\mathrm{A}}^2 C_{\mathrm{Ax}'} & = \rho_{\mathrm{H}} S_{\mathrm{H}} V_{\mathrm{RH}}^2 C_{\mathrm{H}} \cos\beta \\ \rho_{\mathrm{A}} S_{\mathrm{y}'} V_{\mathrm{A}}^2 C_{\mathrm{Ay}'} & = \rho_{\mathrm{H}} S_{\mathrm{H}} V_{\mathrm{RH}}^2 C_{\mathrm{H}} \sin\beta. \end{cases} \tag{7}$$

Dividing the equations in (7) gives

$$\frac{S_{\mathrm{y}'}}{S_{\mathrm{x}'}} \frac{C_{\mathrm{Ay}'}}{C_{\mathrm{Ax}'}} = \tan\beta \tag{8}$$





while taking the sum of squares of the equations in (7), then simplifying and taking the root gives

$$
\quad V_{\mathrm{RH}}^2 = \frac{\rho_A V_A^2 \sqrt{(S_{x'} C_{Ax'})^2 + (S_{y'} C_{Ay'})^2}}{\rho_H S_H C_H}
$$

$$
V_{\mathrm{RH}} = \lambda V_A, \text{ where } \lambda = \sqrt{\frac{\rho_A \sqrt{(S_{x'} C_{Ax'})^2 + (S_{y'} C_{Ay'})^2}}{\rho_H S_H C_H}}. \tag{9}
$$

We have found an expression for the coefficient $\lambda$ (shape parameter), where previous studies used constant values. We see that $\lambda$ is based on the densities, Bluebottle areas and force coefficients. Note that $\lambda$ is also dependent on the angle of attack $\beta_a$, since this affects the value of the force coefficients.

Once $\beta$ and $V_{\mathrm{RH}}$ are known from Eq. (8) and Eq. (9), we calculate the velocity and course of the Bluebottle in terms of the x and y axes. The x and y components of $\mathbf{V}_{\mathrm{RH}}$, the relative velocity of the current, are

$$
\begin{cases}
V_{\mathrm{RHx}} & = V_{\mathrm{RH}} \cos\alpha \\
V_{\mathrm{RHy}} & = V_{\mathrm{RH}} \sin\alpha
\end{cases} \tag{10}
$$

where $\alpha$ is the angle between the x axis and the relative current vector. Relating $\alpha$ to known angles (Fig. 2) gives

$$
\alpha = \beta - \beta_a + \beta_w, \text{ for a left-handed Bluebottle} \tag{11}
$$

$$
\quad \alpha = -\beta - \beta_a + \beta_w, \text{ for a right-handed Bluebottle} \tag{12}
$$

where $\beta_w$ is the angle of the wind (between the x axis and the wind), and $\beta_a$ is the angle of attack (between the x' axis and the wind, Fig. 2) and $\beta$ is the angle between the x' axis and the relative velocity of the current (Fig. 3). Note that $\beta$ is always considered positive. To find $\alpha$ we choose a constant value for $\beta_a$ (see Sect. 5 and Fig. 9).

Once $\alpha$ is known, we find the velocity of the Bluebottle relative to the x and y axes using equations 4 and 10.

$$
\quad \begin{cases}
V_{\mathrm{bbx}} & = u - V_{\mathrm{RH}} \cos\alpha \\
V_{\mathrm{bby}} & = v - V_{\mathrm{RH}} \sin\alpha
\end{cases} \tag{13}
$$

Finally using equation 9:

$$
\begin{cases}
V_{\mathrm{bbx}} & = u - \lambda V_A \cos\alpha \\
V_{\mathrm{bby}} & = v - \lambda V_A \sin\alpha
\end{cases} \tag{14}
$$

Here we can see the Bluebottle velocity is a linear combination of the current velocity and wind velocity with some scaling and rotation. However, the velocity is non-linear with respect to the angle of attack, since both $\alpha$ and $\lambda$ are dependent on $\beta_a$.






This solution for the Bluebottle's velocity can be expressed in a generalised vector form. For a left-handed Bluebottle, we have

$$\mathbf{V}_{\mathrm{bb}} = \mathbf{u} - \lambda V_{\mathrm{A}} \begin{pmatrix} \cos\alpha \\ \sin\alpha \end{pmatrix}, \quad \text{with} \quad \mathbf{u} = (u, v)$$

$$= \mathbf{u} - \lambda V_{\mathrm{A}} \begin{pmatrix} \cos(\beta - \beta_{\mathrm{a}} + \beta_{\mathrm{w}}) \\ \sin(\beta - \beta_{\mathrm{a}} + \beta_{\mathrm{w}}) \end{pmatrix}$$

$$= \mathbf{u} - \lambda V_{\mathrm{A}} \begin{pmatrix} \cos(\beta - \beta_{\mathrm{a}})\cos\beta_{\mathrm{w}} - \sin(\beta - \beta_{\mathrm{a}})\sin\beta_{\mathrm{w}} \\ \sin(\beta - \beta_{\mathrm{a}})\cos\beta_{\mathrm{w}} + \cos(\beta - \beta_{\mathrm{a}})\sin\beta_{\mathrm{w}} \end{pmatrix}$$

Now using the x and y components of the wind velocity, $V_{\mathrm{Ax}} = V_{\mathrm{A}}\cos\beta_{\mathrm{w}}$ and $V_{\mathrm{Ay}} = V_{\mathrm{A}}\sin\beta_{\mathrm{w}}$, we have

$$\mathbf{V}_{\mathrm{bb}} = \mathbf{u} - \lambda V_{\mathrm{Ax}} \begin{pmatrix} \cos(\beta - \beta_{\mathrm{a}}) \\ \sin(\beta - \beta_{\mathrm{a}}) \end{pmatrix} - \lambda V_{\mathrm{Ay}} \begin{pmatrix} -\sin(\beta - \beta_{\mathrm{a}}) \\ \cos(\beta - \beta_{\mathrm{a}}) \end{pmatrix}$$

$$= \mathbf{u} - \lambda \begin{pmatrix} \cos(\beta - \beta_{\mathrm{a}}) & -\sin(\beta - \beta_{\mathrm{a}}) \\ \sin(\beta - \beta_{\mathrm{a}}) & \cos(\beta - \beta_{\mathrm{a}}) \end{pmatrix} \begin{pmatrix} V_{\mathrm{Ax}} \\ V_{\mathrm{Ay}} \end{pmatrix}$$

$$= \mathbf{u} - \lambda \begin{pmatrix} \cos(\beta - \beta_{\mathrm{a}}) & -\sin(\beta - \beta_{\mathrm{a}}) \\ \sin(\beta - \beta_{\mathrm{a}}) & \cos(\beta - \beta_{\mathrm{a}}) \end{pmatrix} \mathbf{V}_{\mathrm{A}}.$$

Using the trigonometric identities $\cos\theta = -\cos(180° - \theta)$ and $\sin\theta = \sin(180° - \theta)$ we have

$$\mathbf{V}_{\mathrm{bb}} = \mathbf{u} + \lambda \begin{pmatrix} \cos(180° - \beta + \beta_{\mathrm{a}}) & \sin(180° - \beta + \beta_{\mathrm{a}}) \\ -\sin(180° - \beta + \beta_{\mathrm{a}}) & \cos(180° - \beta + \beta_{\mathrm{a}}) \end{pmatrix} \mathbf{V}_{\mathrm{A}}$$

And finally:

$$\mathbf{V}_{\mathrm{bb}} = \mathbf{u} + \lambda \mathbf{R}(180° - \beta + \beta_{\mathrm{a}})\mathbf{V}_{\mathrm{A}} \tag{15}$$

where $\mathbf{R}$ represents the rotation matrix. Similarly, for a right-handed Bluebottle we have

$$\mathbf{V}_{\mathrm{bb}} = \mathbf{u} + \lambda \mathbf{R}(180° + \beta + \beta_{\mathrm{a}})\mathbf{V}_{\mathrm{A}} \tag{16}$$

These results show that the velocity of the Bluebottle is a linear combination of the ocean current velocity vector $\mathbf{u}$ and the wind velocity vector scaled by the shape parameter $\lambda$ and multiplied by a clockwise rotation matrix of $180° - \beta + \beta_{\mathrm{a}}$. This can be interpreted as the Bluebottle simply drifting with the current, while the force imparted by the wind on the sail is a proportion ($\lambda$) of the wind velocity at an angle of $180° - \beta + \beta_{\mathrm{a}}$ clockwise from the wind direction.

We refer to equations 15 and 16 as a generalised vector form. By adding assumptions, we can simplify our form to match Lagrangian models seen in previous literature. Hence we can explain the assumptions required to use these previous models. This vector form is the link between the practical papers that used simple vector models (Ferrer and Pastor-Rollan (2017), Ferrer and González (2020), Prieto et al. (2015)) and the theoretical bottom-up approach used by Iosilevskii and Weihs (2009).





## 4.2 Determining Parameters

The formulation shown in equations 15 and 16 for the velocity of the Bluebottle depends on several parameters. Firstly, we will consider the parameters required to find the shape parameter $\lambda$ (see equation 9).

The areas required were measured from photographs taken at Coogee beach, Sydney on 23/01/2019. $S_{y'}$ (area of the Bluebottle's sail on the $y'$ axis) was measured by estimating the area as a segment of a circle. This gave a value of $3\pi - \frac{9\sqrt{3}}{4}$ cm$^2$ (approx $5.5$ cm$^2$). $S_{x'}$ (area of the Bluebottle's sail on the $x'$ axis) was measured by estimating the area as a triangle. This gave 195 a value of $1.12$ cm$^2$.

The force coefficients $C_{Ax'}$ and $C_{Ay'}$ depend on the orientation of the Bluebottle relative to the wind, which is measured by the angle of attack $\beta_a$. $C_{Ax'}$ does not need to be estimated accurately because its value has little effect on the Bluebottle's course and speed, since the product $S_{x'}C_{Ax'}$ (used in equations 8 and 9) is O$(10^{-5})$ while $S_{y'}C_{Ay'}$ is O$(10^{-4})$. Hence for $C_{Ax'}$ we use the same constant value as Iosilevskii and Weihs (2009) of 0.1. $C_{Ay'}$ is calculated using an expression derived by 200 Iosilevskii and Weihs (2009) which estimates $C_{Ay'}$ as a function of angle of attack (Sect. 4.3). Sect. 5 presents examples with different angles of attack and the resulting effect on the Bluebottle's course and speed.

We assume the submerged body of the Bluebottle is a cylinder, so $S_H$ will be the rectangular cross section made by cutting through the cylinder's diameter. This is measured as $3.78$ cm$^2$. Note the tentacles are not taken into account since they can retract or vary their angle. $C_H$ can be estimated as the drag coefficient of a cylinder, which is dependent on Reynolds number. 205 Reynolds number (for sea water) is defined as

$$\mathrm{Re} = \frac{\rho_H u L}{\mu} \tag{17}$$

where $u$ is the current speed with respect to the Bluebottle (order 0.1-1m s$^{-1}$), $L$ is the characteristic length dimension (simply the diameter for a cylinder, which is 0.027m), and $\mu$ is the dynamic viscosity of sea water (order $10^{-3}$ Pa s, dependent on temperature). This gives a Reynolds number between 2781 and 27810 (turbulent flow). For these values, the drag coefficient 210 of a cylinder is very close to 1, so we estimate $C_H$ as 1.

## 4.3 Influence of angle of attack: force coefficient

Iosilevskii and Weihs (2009) derive the following expression for the force coefficient $C_{Ay'}$ as a function of angle of attack by considering the aerodynamic forces acting on a wing surface (slender sail theory).

$$C_{Ay'} = \frac{\pi A}{2}\beta_a + \frac{4\pi A}{3}\frac{f_0}{c}$$
$$A = \frac{b^2}{S_{y'}}$$

where $A$ is the aspect ratio of the sail (calculated using $b$, the sail height, and $S_{y'}$, the sail area), $\beta_a$ is the angle of attack (see Fig. 2), $f_0$ is the sail camber and $c$ is the sail chord (Fig. 4). $f_0/c$ is referred to as the camber ratio.




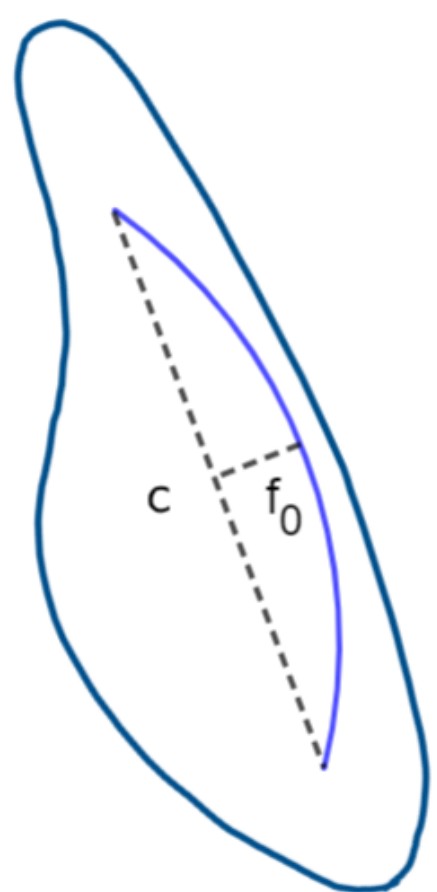

**Figure 4.** Camber of a BB sail. $f_0$ is the sail camber and $c$ is the sail chord

Using our Bluebottle measurements, we have an aspect ratio of roughly 0.35 (half the value used by Iosilevskii and Weihs (2009), whose estimates were based on the PMW) and a sail chord of 5.2cm.

However, there are two concerns with using this expression. Firstly, in aerodynamic theory (for example Abbott et al. (1945)), it has been found that the lift coefficient of an airfoil increases roughly linearly with angle of attack until the lift coefficient approaches a maximum. This behaviour is not considered in this expression.

Secondly, Totton and Mackie (1960) reported that in winds stronger than force 4 (i.e. over 8m/s), the PMW would sail straight downwind with its sail parallel to the wind direction (angle of attack of zero). Our model results do not take this

observation into account if we use the above expression for $C_{\mathrm{Ay}'}$ with 1% camber (the camber value used by Iosilevskii and Weihs (2009)). Instead of a course straight downwind for strong winds, we calculate a Bluebottle course of about $36°$ from the wind. This discrepancy could be solved by ignoring the observation, since it was a second-hand visual observation by a





single individual. However, another explanation is perhaps that the camber ratio is far smaller than Iosilevskii and Weihs (2009) estimated. This is possible since their estimate of 1% was arbitrary.

Despite these concerns, for now this expression remains to be our best method for estimating the value of the force coefficient $C_{\mathrm{Ay}'}$ and will be used throughout this study.

### 4.4    Influence of angle of attack: overall course

In this section we discuss two key parameters: the angle of attack and the sail camber. These are related to the aerodynamic force, so we will assume there is no current and examine how changing the angle of attack varies the Bluebottle's course relative

to the wind. Firstly, a change in the angle of attack means the orientation of the Bluebottle relative to the wind is changing. This results in a different course relative to the wind. Secondly, the force coefficient $C_{\mathrm{Ay}'}$ changes based on the angle of attack, which then varies the Bluebottle's course (Equation 8).

Furthermore, the effect of the angle of attack on the Bluebottle's course is dependent on the camber of the sail (Fig. 5). However, Fig. 5 shows that the camber is only relevant at low angles of attack (0° to 10°). At higher angles of attack the

relationship between angle of attack and the Bluebottle course relative to the wind becomes linear. This is because $C_{\mathrm{Ay}'}$ has become sufficiently large and hence the change in $C_{\mathrm{Ay}'}$ is insignificant. At this point, a 10° change in the Bluebottle's orientation simply results in a 10° change in the Bluebottle's course. As the camber becomes large, $C_{\mathrm{Ay}'}$ becomes sufficiently large even at zero angle of attack, and the relationship tends towards linearity.

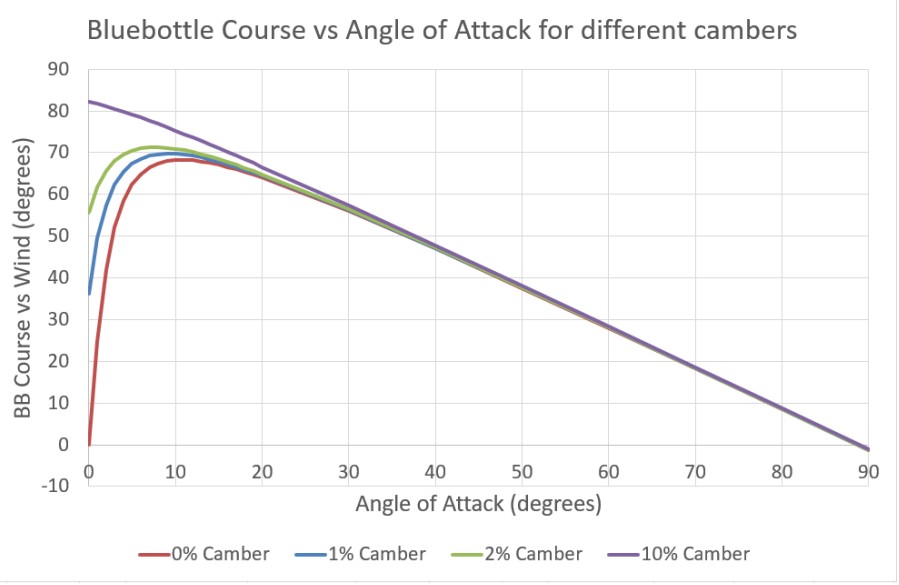

**Figure 5.** Bluebottle course relative to the wind vs angle of attack for different sail camber ratios.

At an angle of attack of zero, the Bluebottle is drifting with its sail parallel to the wind direction and a slight change in

camber will determine the course (Fig. 6). If there is 0% camber the wind will only hit the side of the Bluebottle's sail and





the Bluebottle's course will also be parallel to the wind direction. However, if there is camber the Bluebottle will be pushed sideways (perpendicular to sail) even at zero angle of attack. Due to this, as the camber increases the course tends towards being perpendicular with the wind at zero angle of attack.

At an angle of attack of $90°$ the Bluebottle's sail is perpendicular to the wind direction. We see that, at any camber the
Bluebottle's course is parallel to the wind direction and perpendicular to its sail (Fig. 7).

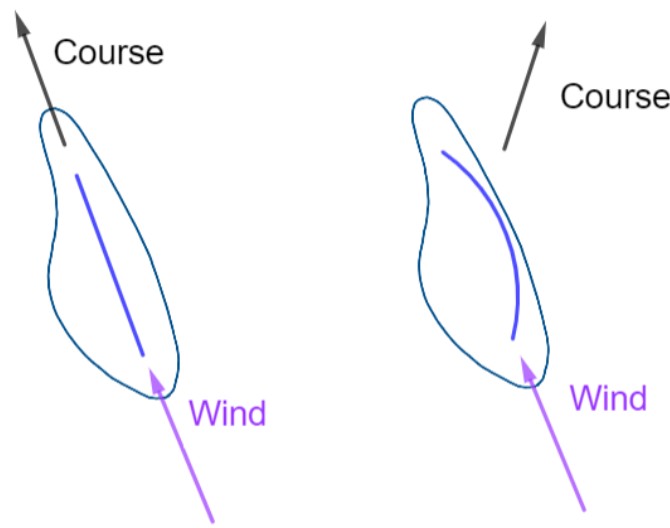

**Figure 6.** Left: Bluebottle sail with zero camber. At zero angle of attack, the Bluebottle will sail downwind with sail parallel to the wind direction. Right: Bluebottle sail with camber. Due to the curvature of the sail, even at zero angle of attack the Bluebottle course will diverge from the wind direction.

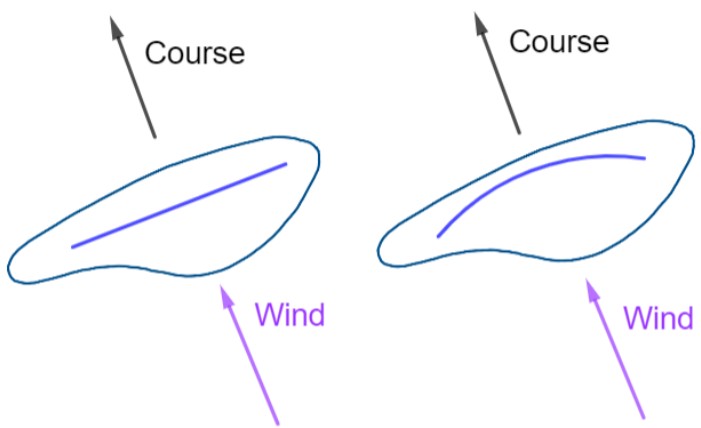

**Figure 7.** At an angle of attack of $90°$, the Bluebottle will sail downwind with sail perpendicular to the wind direction. This occurs at any camber. However, this Bluebottle orientation has not been observed.



## 5 Special Cases

### 5.1 Model and Assumptions for Downwind Drift

We now discuss the assumptions required to reduce our generalised vector form to simpler models seen in some previous studies. Firstly, we must assume the Bluebottle drifts straight downwind with its sail parallel to the wind direction under all

conditions; despite the observation by Totton and Mackie (1960) that the PMW sails in this manner only in winds stronger than force 4 (i.e. over 8m/s). Considering a downwind drift, the sail orientation corresponds to an angle of attack $\beta_a$ of 0° (Fig. 6). Substituting this into our vector form (Equation 15) gives

$$\mathbf{V}_{bb} = \mathbf{u} + \lambda \mathbf{R}(180° - \beta)\mathbf{V}_A$$

Now, the Bluebottle will only sail straight downwind if the force coefficient $C_{Ay'}$ equals zero, resulting in no aerodynamic

force perpendicular to the sail (Fig. 8). At an angle of attack of zero, we expect $C_{Ay'}$ to equal zero only if there is no camber (Fig. 5). Hence we must also make this assumption that the sail camber is zero. Based on equation 8 and the fact that the relative current will always be opposite the wind (in order to have an equilibrium condition), we have $\beta = 180°$ (Fig. 8), and $\mathbf{R}$ is the identity matrix. Using Equation 16 also gives this result. This gives a form that we have seen in previous literature (discussed in Sect. 6).

$$\mathbf{V}_{bb} = \mathbf{u} + \lambda \mathbf{V}_A \qquad (18)$$

We can also exclude the current by assuming $\mathbf{u} = 0$ to reach another form seen in previous literature.

$$\mathbf{V}_{bb} = \lambda \mathbf{V}_A \qquad (19)$$

Hence we see that the models used in previous literature can be verified by our generalised form, but require fairly strong assumptions of downwind course in all conditions and no camber. In Sect. 5.2 and 5.3 we lift these assumptions to further

explore the drifting dynamics of the Bluebottle.

It should be noted that, according to our model, a course straight downwind can also occur if the Bluebottle has an angle of attack of 90° (Fig. 5 and Fig. 7). However, Totton and Mackie (1960) did not observe angles of attack greater than 40° to 45° in their physical experiments. Hence we assume that the Bluebottle cannot orient at an angle of attack of 90°.

### 5.2 Case with angle of attack of 40° and no current

Totton and Mackie (1960) observed that in light winds the PMW balances itself at approximately 40° to the wind. Hence, we now assume $\beta_a$ has a value of 40° or −40°, depending on whether the Bluebottle is right-handed (left-sailing) or left-handed (right-sailing). Since the orientation of the Bluebottle is constant relative to the wind, the sail is always hit by the wind at the same angle and the force coefficients $C_{Ax'}$ and $C_{Ay'}$ can be considered as constants. For $C_{Ax'}$ we will use the value 0.1 as explained in Sect. 4.2. For $C_{Ay'}$ we use the equation in Sect. 4.3, giving a value of 0.40. Recall that $C_{Ay'}$ varies depending on

the angle of attack. For example, from 0.31 to 0.50 for an angle of attack of 30° to 50°.

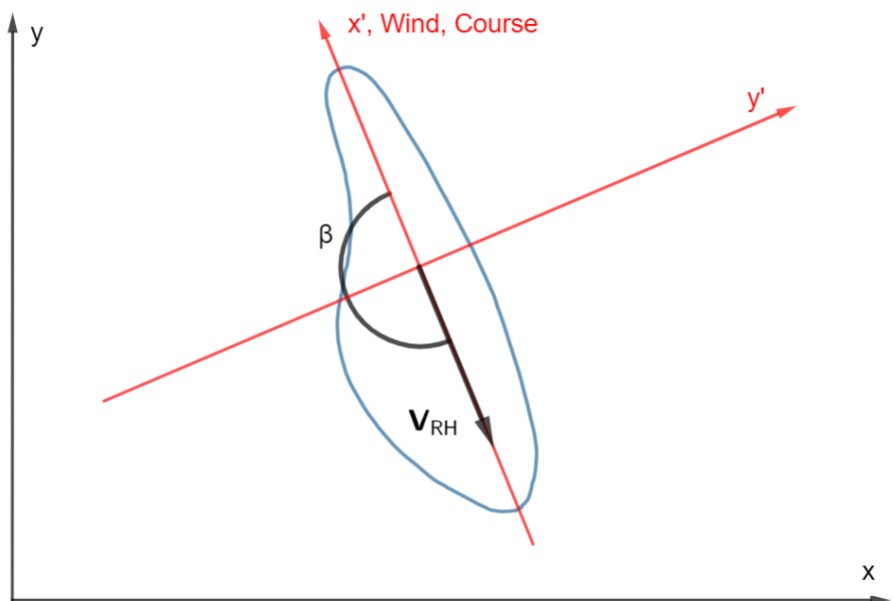

**Figure 8.** Case with no current and Bluebottle sailing straight downwind with sail parallel to the wind direction ($\beta_\mathrm{a} = 0$). Note that the wind direction and BB course are on the $x'$ axis.

Assuming there is no current, our form simplifies to

$$\mathbf{V}_\mathrm{bb} = \lambda \mathbf{R}(180° - \beta + \beta_\mathrm{a})\mathbf{V}_\mathrm{A}$$

$\beta$ can be calculated using equation 8 to be $87.1°$ (varies from $86.2°$ to $87.7°$ for an angle of attack of $30°$ to $50°$), while $\lambda$ is calculated using equation 9 to be $0.0266$ (varies from $0.023$ to $0.030$ for an angle of attack of $30°$ to $50°$). We will consider an

example with a left-handed (right-sailing) Bluebottle, thus $\beta_\mathrm{a} = -40°$. Substituting all these values gives

$$\mathbf{V}_\mathrm{bb} = 0.0266\mathbf{R}(52.9°)\mathbf{V}_\mathrm{A}$$

We now have a clockwise rotation matrix. Hence, this means that the left-handed Bluebottle will drift at an angle of $52.9°$ clockwise (to the right) from the wind at $2.66\%$ of the wind speed. Similarly, a right-handed Bluebottle will drift at an angle of $52.9°$ anticlockwise (to the left) from the wind at $2.66\%$ of the wind speed. Note that since there is no current, the hydrodynamic

force is only the drag from the submerged part of the Bluebottle. Hence the relative current vector $\mathbf{V}_\mathrm{RH}$ is directly opposite the Bluebottle's motion (Fig. 9).





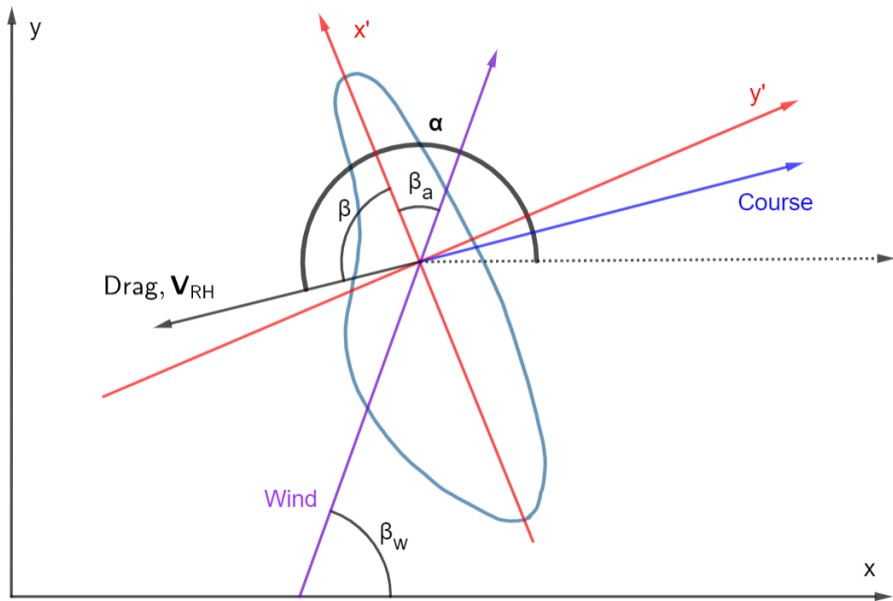

**Figure 9.** Case with no current for a left-handed (right-sailing) Bluebottle ($\beta_a = -40°$). Note that $\beta_a$ is negative because it is the angle between the $x'$ axis and the wind, which is clockwise. Length of Course and $\mathbf{V}_{RH}$ are to scale. Wind vector length has been scaled down by a factor of 8.

### 5.3 Case with angle of attack of $40°$ with current

Including the current does not have any impact on the calculated values of $\beta$ or $\lambda$. Hence, the effect of the wind on the Bluebottle's speed and orientation will remain the same. By adding the current we have

$$\mathbf{V}_{bb} = \mathbf{u} + 0.0266\mathbf{R}(52.9°)\mathbf{V}_A$$

Unlike Sect. 5.2, with the ocean current added the relative current velocity $\mathbf{V}_{RH}$ is no longer directly opposite the Bluebottle's course (Fig. 10). However, since the wind conditions and other variables have not changed from Sect. 5.2, the aerodynamic force is identical. Hence, we also require the relative current vector to be identical since this determines the hydrodynamic force, which must balance out the aerodynamic force in order to have an equilibrium condition. This means that the course of

the Bluebottle must adjust such that $\mathbf{V}_{RH}$ is in the same position as Sect. 5.2 ($\beta = 87.1°$). Fig. 11 shows examples of different current conditions with a constant wind. In each example, $\mathbf{V}_{RH}$ is kept in the same position by adjusting the Bluebottle's course.

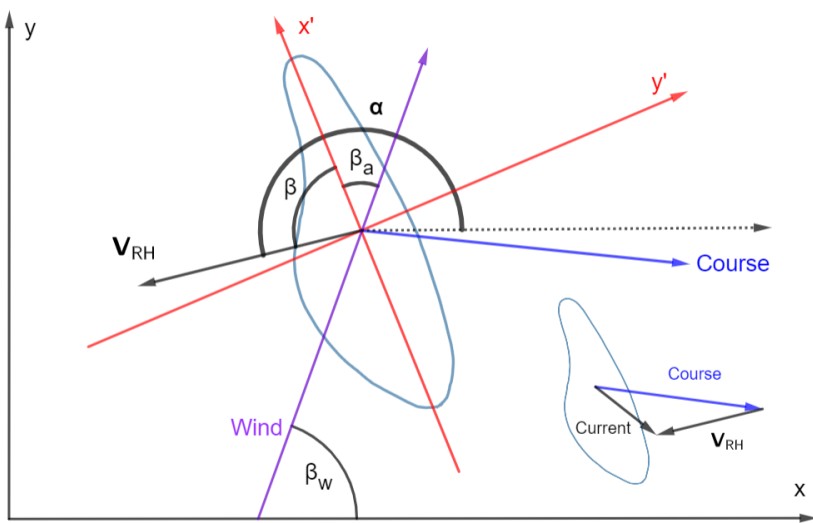

**Figure 10.** Case with current running south-east for a left-handed (right-sailing) Bluebottle ($\beta_a = -40°$). Diagram in the bottom right shows the vector addition between the Bluebottle's course, the current and the relative current felt by the Bluebottle. Length of Course, Current and $\mathbf{V}_{RH}$ are to scale, based on a current speed that is 5% of the wind speed. Wind vector length has been scaled down by a factor of 8.

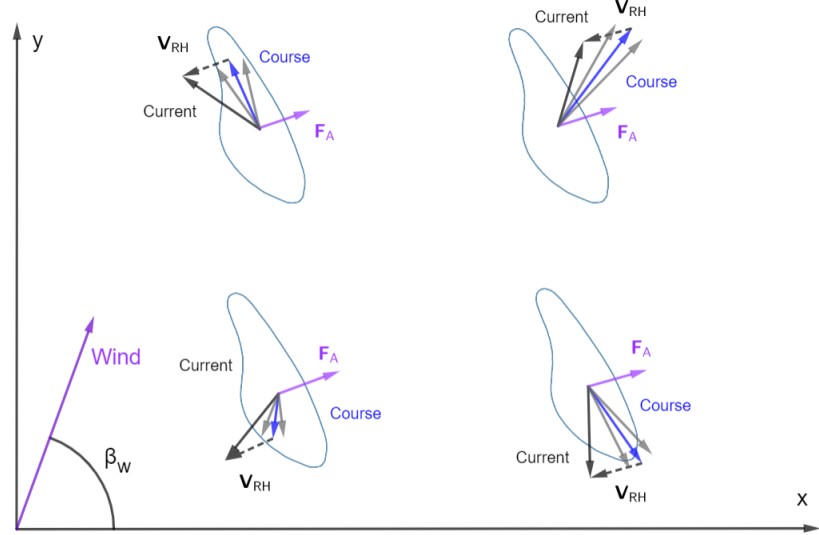

**Figure 11.** Examples of different currents with a constant wind. Grey vectors indicate confidence intervals for the Bluebottle course if we consider that the angle of attack ($\beta_a$) could be 30 to 50°. Note that $\mathbf{V}_{RH}$ must always point in the same direction. Length of Course, Current and $\mathbf{V}_{RH}$ are to scale, based on a current speed that is 5% of the wind speed. Wind vector length has been scaled down by a factor of 8.




## 6 Discussion and Conclusion

We have added the ocean background current to an existing theoretical model, taking the aerodynamic and hydrodynamic

forces acting on the Bluebottle and solving the equilibrium condition to create a generalised vector model for the speed and course of the Bluebottle. Adding assumptions to our generalised model results in simplification to models that have been seen in previous literature. The generalised vector form is the link between the practical papers that used simple Lagrangian vector models and the theoretical papers that used a bottom-up approach. We also identify and discuss the key parameters: shape ($\lambda$), angle of attack and camber. Finally, we find that under typical sailing conditions the Bluebottle will drift at an angle of $52.9°$

from the wind at 2.66% of the wind speed, plus the velocity of the ocean background current.

It is worth noting that the form of equations 15 and 16 are different from leeway methods, which consider the motion of the object as a wind vector plus leeway (divergence from the wind). The leeway is often estimated using physical experiments and statistics. This methodology has been used in many previous studies (Breivik et al. (2011), Hackett et al. (2006), Ni et al. (2010), Wang et al. (2015)). Our theoretical bottom-up approach to calculate Bluebottle drift using force balance in equilibrium

is a completely different methodology, despite the similar looking vector form that results from both techniques.

Ferrer and Pastor-Rollan (2017) modelled the drift velocity of PMWs as wind velocity multiplied by a wind drag coefficient estimated from numerical simulations. The current was not used because wind was considered to be the main mechanism, based on Ferrer et al. (2014) who used symmetric fish tags for model calibration. The model calibration used in Ferrer et al. (2014) included a calculation of surface current as the combination of ROMS (Regional Ocean Modeling System) current and

wind. They also assume that the PMWs drift straight downwind. This matches our form in equation 19, where we assume the Bluebottle has zero angle of attack and there is no current. The model used by Prieto et al. (2015) was advection by surface currents (computed by ROMS model) plus 10% of wind velocity. This matches our form in equation 18, still assuming an angle of attack of zero. In comparison, we have a generalised form for calculating the drift of a Bluebottle for different angles of attack. We use a bottom-up approach for our wind velocity coefficient $\lambda$, which is determined by the specific areas and force

coefficients of the bluebottle. We find that when the bluebottle has an angle of attack of $40°$, as suggested by observations, $\lambda = 0.0266$ and the wind pushes the bluebottle on a course of about $53°$ to the wind. This value of $\lambda$ falls within the range of 0.02-0.045 tested by Ferrer and Pastor-Rollan (2017) but differs from the 10% used by Prieto et al. (2015). The $\alpha$ term in our model accounts for the angle of the current, the angle of the wind and the bluebottle's angle of attack. Our model also incorporates the current, which has a significant effect even at 10m/s winds. Indeed, at wind speeds of O(10), the speed

imparted onto the Bluebottle from the wind is about 0.266m/s (2.66% of wind speed), which is the same order of magnitude as the speed of ocean currents. Hence based on our vector form, the current should not be ignored when predicting the drift of the bluebottle. Note that the current relevant for this study is the surface ocean current which would be felt by the <10cm Bluebottle. Relatively good results from modelling studies which did not take into account the ocean current can be explained by the great impact the wind has on the top few centimeters of the ocean.

It should be noted that the influence of waves is not taken into account in our model. The impact forces of waves has an effect on drifters (Szelangiewicz and Żelazny (2018)), but several additional variables and functions are required to calculate




this impact force. This will make our model significantly more complex, and currently cannot realistically be added. The effect of Stokes drift, however, could be added into the ocean surface current vector, following Clarke and Vander (2018) for instance. Stokes drift is a phenomenon that occurs on the ocean surface where the surface waves affect the net particle movement in
the top 1 or 2m of the ocean, in the direction of the waves. Clarke and Vander (2018) found that Stokes drift is mostly in the direction of the wind, and thus it is mainly due to shorter waves generated by the local wind. It was also found that the magnitude of Stokes drift can be approximated by

$$u_{\text{Stokes}} = 4.4u_* \ln\left(\frac{0.0074u_{10}}{u_*}\right) \tag{20}$$

where $u_{10}$ is the 10m wind speed and $u_*$ is a parameter calculated by

$$u_* = \sqrt{\frac{|\tau_{\text{o}}|}{\rho_{\text{w}}}} \tag{21}$$

where $|\tau_{\text{o}}|$ is the wind stress magnitude and $\rho_{\text{w}}$ is the density of water. Assuming a constant drag coefficient, Clarke and Vander (2018) then show that $u_{\text{Stokes}}$ can be estimated as 1% of $u_{10}$. This Stokes drift estimate can be added as an additional term to equation 15, giving

$$\mathbf{V}_{\text{bb}} = \mathbf{u} + 0.01\mathbf{V}_{\text{A}} + \lambda\mathbf{R}(180° - \beta + \beta_{\text{a}})\mathbf{V}_{\text{A}}. \tag{22}$$

However, another consideration is that large waves (at high wind speeds) may break on top of the Bluebottle and result in the Bluebottle becoming imbalanced or even toppling over. Our model cannot predict the behaviour of the Bluebottle in this situation.

Another assumption in our model is that the body of the Bluebottle is assumed to be a perfectly symmetrical cylinder with no tentacles. In reality, the submerged body is not perfectly symmetrical so the hydrodynamic force would also influence the
orientation of the Bluebottle's sail and body to some extent, like the keel of a sailboat. However, since the Bluebottle can extend and retract the tentacle, changing its length and angle, it is hard to model. The drag force from the Bluebottle's single tentacle (calculated using equations from Iosilevskii and Weihs (2009)) is $\mathrm{O}(10^{-2})$, which is insignificant compared to the overall hydrodynamic force acting on the Bluebottle of $\mathrm{O}(1)$.

This study is focused on the Bluebottles found on the east coast of Australia. Different parameter values may be required
for the larger PMW. For example, Iosilevskii and Weihs (2009) use an estimate of 0.7 for the aspect ratio of the PMW's sail, which affects the force coefficient $C_{\text{Ay}'}$ (Sect. 4.3). This is much larger than our measured aspect ratio for the Bluebottle's sail of 0.35. In Sect. 5.2 we conclude that at an angle of attack of $40°$ and no current, a left-handed Bluebottle will drift at an angle of $52.9°$ clockwise from the wind at 2.66% of the wind speed. A PMW in the same conditions will drift at an angle of $51.5°$ clockwise from the wind at 3.73% of the wind speed. While the drift angle is almost the same, the velocity of the PMW is 40%
higher than the Bluebottle due to the higher aspect ratio leading to a larger force coefficient $C_{\text{Ay}'}$. It is worth noting that the drag caused by the PMW's many long tentacles may affect its velocity significantly more than for the Bluebottle.

Further research that would supplement this study and potentially lead to the prediction of Bluebottle beachings include:



- physical experiments to observe Bluebottle drifting and estimate the key parameter values (e.g. force coefficients, angle of attack, camber) of our model.

- a detailed understanding of the specific habitat and life cycle of the Bluebottle to determine the starting point for a drift model.

We hope our work, creating a generalised model for Bluebottle drift, encourages new research in this area and helps in the future development of a forecasting tool that can prevent tens of thousands of beachgoers from experiencing painful Bluebottle stings.

# 1  Appendix A

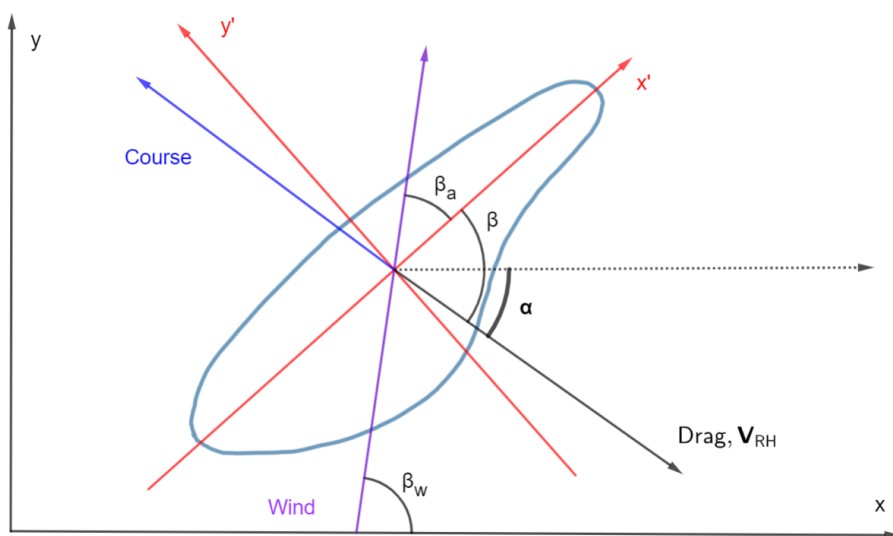

**Figure A1.** Case with no current for a right-handed (left-sailing) Bluebottle ($\beta_a = 40°$) Length of Course and $\mathbf{V}_{RH}$ are to scale. Wind vector length has been scaled down by a factor of 8.

*Author contributions.* D.L. led the writing of the manuscript, completed the derivations and performed the calculations. A.S. contributed to writing the manuscript and the derivations. S.G. provided the idea and helped with derivation and writing of the manuscript.

*Competing interests.* The authors declare that they have no conflict of interest.



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
