# Peer review of "Drifting dynamics of the Bluebottle (*Physalia physalis*)"

_Ocean Science, 2021_

## Referee Comment (RC2)

This article proposes a more complex mathematical model than previous models published in the literature to analyse the possible drift of Bluebottles due to the combination of wind and surface ocean currents. After reading the article, here are some general considerations to take into account in a discussion on this specific topic. In general, the size of the float of the Portuguese man-of-war that we find on the beaches is usually less than 15 cm, their tentacles do not usually exceed 3 metres in length and their wet weight can average 25-30 grams. In other words, we are talking about an extremely light organism, whose float is able to rotate very easily even in low winds. For example, in some parts of this video we can see a Portuguese man-of-war and how easily its float moves even in calm waters:

https://www.eitb.eus/es/eltiempo/videos/detalle/1382860/video-prediccion-llegada-medusas--azti/

Because this organism lives in the atmosphere-sea interface and we can say that the wind is the most important mechanism of its drift, since the surface ocean currents at the top level of the water column and the local waves are generated by the wind. That is, it is expected that the wind variable from prediction meteorological models (used in simple or complex drift models), by itself, will provide very adequate and useful information to model the drift of these organisms in a fairly efficient way. We are corroborating this fact in experiments with the launching of very light (<200 grams of weight) and cylindrical surface drifting buoys of approximately 10 cm x 10 cm (height x width). The trajectories of these symmetrical buoys or parts of these trajectories are adjusted in a very acceptable way with a simple drift model based only on wind (in black the real data and in red the simulated results in the figures below show some adjustment examples). Now we are analysing the drift of non-symmetrical buoys in order to simulate the dimorphism in the Portuguese man-of-war.

[Figure]

What we observe is that the greater the change in the characteristics of the real wind over time, the worse the prediction of this wind and the simulated trajectory of the buoys, since the meteorological model that predicts the wind offers worse results. Obviously, in areas where the current due to the tide is important or there are significant river discharges or the general circulation not induced by the local wind is intense or the swell is important, adding these components to the direct and indirect effect of the wind would be interesting. However, it would be necessary to predict these components with great precision and spatiotemporal resolution in order to obtain an optimal drift model. Currently, the coefficients of determination ($r^2$) of the wind in comparative meteorological model-real data are around a value of 0.7 in open waters and are below 0.5 in very coastal areas. If we take into account that the models of surface ocean currents (in the first centimetres of the water column) and the wave models depend on this

wind prediction, we find that the $r^2$ of these models (currents and waves) decrease significantly, especially in direction. In other words, the fact of adding these components in a complex drift model does not seem a priori to offer a better solution than a simple drift model based only on wind. In addition, the velocities of these components should be also calibrated with coefficients for the specific organism or object being analysed. In fact, comparisons that we have carried out incorporating these components from different numerical current prediction models (HYCOM, NEMO, etc.) do not improve the predictions of the trajectories of the buoys based on a simple drift model that uses only the variable wind from meteorological models. Some recent works with heavier surface drifting buoys (for example, van Sebille et al. (2021), Dispersion of Surface Drifters in the Tropical Atlantic) is showing the need to greatly improve the outputs of leading hydrodynamic models that assimilate real data in order to reproduce the surface drift of objects. However, although it is possible to reasonably model a very specific object, we must be aware that accurately modelling the drift of an organism such as the Portuguese man-of-war from its initial stage to its adult stage or when it ends up on a beach is a task almost impossible. Furthermore, the complexity of a drift model that uses data from various prediction models (atmosphere, ocean and waves) does not guarantee a better solution to the problem. Additionally, complex models are associated with a considerable number of strong assumptions and coefficients or parameters to be determined for each specific organism. That is, they are very specific models for an organism with certain characteristics and cannot be considered as generalized models to solve a problem.

In the complex physical and mathematical approach presented in this article (I assume it is only applicable to an adult organism), the first thing that should be analysed is where the centre of gravity of the organism possibly is in order to correctly draw the diagram of forces. It is expected that the greatest weight of the organism is in the part of its tentacles (and more when the organism has trapped food), for which the centres of gravity of left-handed and right-handed individuals would have to be at different points. On the other hand, the Portuguese man-of-war is not symmetrical and assuming a perfect symmetry or that the submerged part where the tentacles are is a cylinder (and other assumptions discussed throughout the article) greatly conditions the results obtained. This is why the mathematical results obtained by the authors should be examined with great caution.

For example, in Figure 7, you write the following: "Figure 7. At an angle of attack of 90, the Bluebottle will sail downwind with sail perpendicular to the wind direction. This occurs at any camber. However, this Bluebottle orientation has not been observed". If we take into account that the Portuguese man-of-war is asymmetric and its greatest weight is in the area of the tentacles, given the wind situation shown in figure 7, it would be reasonable to think that the front part of the organism would move faster than the part rear by rotating the organism clockwise to achieve a balanced position and changing the orientation of the float relative to the wind. The same happens in Figures 10 and 11 in which a current is introduced. Considering that the combined effect of wind and surface current will change the course of the organism, but it will not change the orientation of the float with respect to the wind seems to me a very strong assumption. I fully understand that trying to mathematically model the drift of this organism is extremely complicated and the authors have made a significant effort to do so. In my opinion, it would be interesting to analyse how the orientation of the float changes in conditions of wind and surface current, especially when the direction of the surface currents is far away from the direction of the wind or from the direction obtained assuming that Ekman's theory is valid and it is perfectly fulfilled (that is, surface current at 45 degrees with respect to the wind). Possibly this can only be done in controlled experiments in a wave-current tank.

Please, substitute along the paper the following reference "**Ferrer and Pastor-Rollan (2017)**" by "**Ferrer and Pastor (2017)**".

---

## Author Response (AR1)

**1 Point-by-point response to RC1 (Laura Prieto)**

**Reviewer:**
The work presented in this manuscript is a step forward in the theory physical mechanics of how the colonies of the hydrozoan Physalia physalis are affected by the surface currents and surface winds. The authors performed a well written study, taking into account the main references regarding these organisms and comparing the parameters values obtained in this study with other studies (more experimental) The good part is that both in the analysis and in the discussion of their results, they are aware of the big differences in hydrodynamics due to the simple effect of the size (much larger) of the Atlantic Portuguese Man of War compare to the Indo-Pacific Bluebottle (the focus of this paper).

Another fact that I was missing while I was reading the manuscript, but that the authors finally mentioned at the end of the text, is which should be the next step in their study that is to perform some real experiments to verify the parameters obtained in the theory model developed with the Bluebottle. Without these experiments, this manuscript will be just a nice theory model, without validation for these small colonies. Regarding the habitat, of course it would be very interesting to know more about it, and it could be connected with the point just mentioned.

**Author:**
We thank the reviewer for their comments. We agree that physical experiments are required to validate the model and determine key parameter values as for any theoretical model. We plan to complete this research in the future but this is beyond the scope of the current study.

**Author's changes:**
None.

**Reviewer:**
Line 21: correct to pneumatophore. Lines 325-329: several times, correct to Bluebottle (to be coherent to the rest of the text).

**Author:**
We have corrected the spelling error on line 21. Regarding lines 325-329, referencing the PMW rather than the Bluebottle is intentional here. We use the term PMW when we discuss previous research which was done specifically for the larger Atlantic variant.

**Author's changes:**
Corrected spelling error (line 21 in track-changes file).

**2 Point-by-point response to RC2 (Luis Ferrer)**

**Reviewer:**

30 This article proposes a more complex mathematical model than previous models published in the literature to analyse the possible drift of Bluebottles due to the combination of wind and surface ocean currents. After reading the article, here are some general considerations to take into account in a discussion on this specific topic. In general, the size of the float of the Portuguese man-of-war that we find on the beaches is usually less than 15 cm, their tentacles do not usually exceed 3 metres

35 in length and their wet weight can average 25-30 grams. In other words, we are talking about an extremely light organism, whose float is able to rotate very easily even in low winds. For example, in some parts of this video we can see a Portuguese man-of-war and how easily its float moves even in calm waters:

https://www.eitb.eus/es/eltiempo/videos/detalle/1382860/video-prediccion-llegada-medusas–azti/

Because this organism lives in the atmosphere-sea interface and we can say that the wind is the most important mechanism

40 of its drift, since the surface ocean currents at the top level of the water column and the local waves are generated by the wind. That is, it is expected that the wind variable from prediction meteorological models (used in simple or complex drift models), by itself, will provide very adequate and useful information to model the drift of these organisms in a fairly efficient way. We are corroborating this fact in experiments with the launching of very light (<200 grams of weight) and cylindrical surface drifting buoys of approximately 10 cm x 10 cm (height x width). The trajectories of these symmetrical buoys or parts of these

45 trajectories are adjusted in a very acceptable way with a simple drift model based only on wind (in black the real data and in red the simulated results in the figures below show some adjustment examples). Now we are analysing the drift of non-symmetrical buoys in order to simulate the dimorphism in the Portuguese man-of-war.

**Author:**

We thank the reviewer for providing an interesting and engaging discussion. We have enjoyed reading their comments and

50 watching the video, and will include some of these points in our manuscript. We are looking forward to seeing the results of the asymmetric drifter experiments, and agree that this is the way forward to validate Lagrangian particle tracking models, provided that the buoys behave similarly to Physalia.

We thank the reviewer's for the comment on the size of the Portuguese Man of War, and have amended the text to "up to 15 cm". While we agree that that tentacles most likely don't exceed 3 metres when not fully extended, Munro et al. (2019)

55 mention that they can reach up to 30 m in mature colonies. We will amend this as well.

**Author's changes:**

Amended lengths and added Munro et al. as reference (line 32 in track-changes file).

**Reviewer:**

60 What we observe is that the greater the change in the characteristics of the real wind over time, the worse the prediction of this wind and the simulated trajectory of the buoys, since the meteorological model that predicts the wind offers worse results. Obviously, in areas where the current due to the tide is important or there are significant river discharges or the general circulation not induced by the local wind is intense or the swell is important, adding these components to the direct and indirect effect of the wind would be interesting. However, it would be necessary to predict these components with great precision and

65 spatiotemporal resolution in order to obtain an optimal drift model. Currently, the coefficients of determination (r2 ) of the wind in comparative meteorological model-real data are around a value of 0.7 in open waters and are below 0.5 in very coastal areas. If we take into account that the models of surface ocean currents (in the first centimetres of the water column) and the wave models depend on this wind prediction, we find that the r2 of these models (currents and waves) decrease significantly, especially in direction. In other words, the fact of adding these components in a complex drift model does not seem a priori to

70 offer a better solution than a simple drift model based only on wind. In addition, the velocities of these components should be also calibrated with coefficients for the specific organism or object being analysed. In fact, comparisons that we have carried out incorporating these components from different numerical current prediction models (HYCOM, NEMO, etc.) do not improve the predictions of the trajectories of the buoys based on a simple drift model that uses only the variable wind from meteorological models. Some recent works with heavier surface drifting buoys (for example, van Sebille et al. (2021), Dispersion of

75 Surface Drifters in the Tropical Atlantic) is showing the need to greatly improve the outputs of leading hydrodynamic models that assimilate real data in order to reproduce the surface drift of objects. However, although it is possible to reasonably model

a very specific object, we must be aware that accurately modelling the drift of an organism such as the Portuguese man-of-war from its initial stage to its adult stage or when it ends up on a beach is a task almost impossible. Furthermore, the complexity of a drift model that uses data from various prediction models (atmosphere, ocean and waves) does not guarantee a better solution to the problem. Additionally, complex models are associated with a considerable number of strong assumptions and coefficients or parameters to be determined for each specific organism. That is, they are very specific models for an organism with certain characteristics and cannot be considered as generalized models to solve a problem.

**Author:**

The reviewer starts by discussing the effectiveness of modelling drift on the ocean surface using wind only. One argument for this strategy is that surface currents in the top few centimeters are mostly winds-driven, and not accurate in ocean models. While this is a good point, we note that the scientific community is increasingly interested in understanding the surface current shear (e.g. Haza., 2019), and the drifter experiments mentioned above go in the same direction. In our discussion we note that "the current relevant for this study is the surface ocean current which would be felt by the Bluebottle. Relatively good results from modelling studies which did not take into account the ocean current can be explained by the great impact the wind has on the top few centimeters of the ocean." (lines 332-334). We acknowledge that a model based only on wind can be an effective simplified solution to estimate drift. However, as mentioned by the reviewer, ocean currents will be critical during low winds or in situations where the current and wind have a similar effect on the Bluebottle. In addition, pre-existing ocean circulation was shown to be significant (although not dominant) for the drift of surface drifters in the top 5cm of the ocean during strong wind events (Lodise et al., 2019). The purpose of our manuscript is to present a theoretical model, based on the aerodynamic and hydrodynamic forces acting on the Bluebottle, as a foundation for improved modelling and future research. This is explicitly written lines 57-58 and 10-11: "Here we expand that model by adding the effect of the ocean current and simplifying the model down to an intuitive generalised vector form which could be implemented in Lagrangian models". "Adding assumptions to this generalised model allows us to retrieve models used in previous literature". The existence of wind based modelling methods does not diminish our interest in a theoretical modelling method.

**Author's changes:**

None.

**Reviewer:**

In the complex physical and mathematical approach presented in this article (I assume it is only applicable to an adult organism), the first thing that should be analysed is where the centre of gravity of the organism possibly is in order to correctly draw the diagram of forces. It is expected that the greatest weight of the organism is in the part of its tentacles (and more when the organism has trapped food), for which the centres of gravity of left-handed and right-handed individuals would have to be at different points. On the other hand, the Portuguese man-of-war is not symmetrical and assuming a perfect symmetry or that the submerged part where the tentacles are is a cylinder (and other assumptions discussed throughout the article) greatly conditions the results obtained. This is why the mathematical results obtained by the authors should be examined with great caution.

**Author:**

Another point of discussion refers to some of the assumptions used in our study. At the beginning of section 3 (page 5), we discuss the aerodynamic and hydrodynamics centres of effort used by Iosilevskii and Weihs (2009) and the reasons we exclude these from our model. Currently, we do not have the specific measurements or controlled experiments required to analyse the centre of effort, nor to analyse the way Physalia moves as the wind and current conditions change, while considering its asymmetry. We state (lines 98-99) "similarly to the leeway methodology (discussed further in Sect. 6), the orientation behaviour of the Bluebottle (mainly represented by the angle of attack) can instead be determined by observations made in physical experiments". Once this information is available, it will be easily integrated in our model. Regarding the tentacles, the Bluebottles found on the east coast of Australia (which are the focus of this study) only have one main tentacle that is far shorter than that of the Portuguese Man-of-War found in the Atlantic, thus making the tentacles far less significant. The Bluebottle we photographed on Coogee beach, Sydney had a tentacle with a diameter of 1mm and length of less than 40cm when extended, and the area if more important than the weight to determine the physical forces horizontally. In our discussion (lines 356-358) we state that "The drag force from the Bluebottle's single tentacle (calculated using equations from Iosilevskii and Weihs 2009) is $O(10-2)$, which is insignificant compared to the overall hydrodynamic force acting on the Bluebottle of $O(1)$." Also, our photographs of the Bluebottle in water show the submerged body is a clump underneath the float with a similar width at many

different angles, so the shape can be reasonably estimated as a symmetric cylinder under the water. Finally, the mathematical results in Section 5 provide case studies with simplified hypothesis in order to illustrate the results of the rather complicated model, by demonstrating its use and relating the results to previous modelling studies. As explained in the discussion (page 20), future research focusing on physical experiments are required to determine key parameter values and a detailed understanding of the specific habitat and life cycle of the Bluebottle.

**Author's changes:**
None.

**Reviewer:**
For example, in Figure 7, you write the following: "Figure 7. At an angle of attack of 90, the Bluebottle will sail downwind with sail perpendicular to the wind direction. This occurs at any camber. However, this Bluebottle orientation has not been observed". If we take into account that the Portuguese man-of-war is asymmetric and its greatest weight is in the area of the tentacles, given the wind situation shown in figure 7, it would be reasonable to think that the front part of the organism would move faster than the part rear by rotating the organism clockwise to achieve a balanced position and changing the orientation of the float relative to the wind.

**Author:**
Regarding the reviewer's comments on Figure 7 from Section 4.4, the purpose of this section is to analyse how adjusting the values of angle of attack and camber influence the Bluebottle's course. Since there are previous studies that model Physalia's drift as straight downwind, we include Figures 6 and 7 to show which model inputs result in a course straight downwind, and to demonstrate the capacity of the model to represent dynamics previously observed. Figures 6 and 7 are showing a Bluebottle with an angle of attack that is assumed to be a constant value of 0 and 90° respectively, which are the two ways to achieve this downwind course. However, this does not mean we believe both orientations would occur in the real world. We agree that in the situation shown in Figure 7, the Bluebottle would rotate to achieve a balanced position, thus changing its orientation. As we mention, a constant angle of attack of 90° has not been observed. We will remove this figure as it can be misleading and will rewrite the section to clarify this.

**Author's changes:**
Removed misleading figure and improved explanation (lines 245-269 in track-changes file). This resulted in previous information being repeated, so the structure of previous sections was changed (lines 195-244 in track-changes file).

**Reviewer:**
The same happens in Figures 10 and 11 in which a current is introduced. Considering that the combined effect of wind and surface current will change the course of the organism, but it will not change the orientation of the float with respect to the wind seems to me a very strong assumption. I fully understand that trying to mathematically model the drift of this organism is extremely complicated and the authors have made a significant effort to do so. In my opinion, it would be interesting to analyse how the orientation of the float changes in conditions of wind and surface current, especially when the direction of the surface currents is far away from the direction of the wind or from the direction obtained assuming that Ekman's theory is valid and it is perfectly fulfilled (that is, surface current at 45 degrees with respect to the wind). Possibly this can only be done in controlled experiments in a wave-current tank.

**Author:**
Finally, the reviewer questions Figures 10 and 11 and the orientation assumptions. As stated above, the purpose of the cases (which use a constant value for angle of attack) is to demonstrate the use of the model in a simplified case of a constant angle of attack, which is the main parameter controlling the orientation of the Bluebottle. We choose to use the most reasonable constant values of angle of attack (40° as observed by Totton and Mackie, 1960) for our examples, rather than defining an arbitrary function for angle of attack. Again, we agree with reviewer, controlled experiments in a tank or in-situ surveys will be the best way to determine how the angle of attack changes with conditions. Once this is achieved, the parameter of our model will easily be implemented, as a constant value or as a function of wind speed for example.

**Author's changes:**
Added discussion about orientation assumption (lines 380-485 in track-changes file).

**3 Other changes**

175 – Addition of "Physalia Physalis" in title.

– Added reference to Shannon and Chapman (1983): lines 50-52 in track-changes file.

– Various improvements to figures: addition of arrows to indicate angle direction where necessary, some changes to captions, and Figure 6 changed to panels.

– Various minor changes to fix errors, consistency, grammar and clarity.

---

## Author Response (AR2)

**1   Response to Editor (Piers Chapman)**

**Editor:**
The authors have revised their paper and it is suitable for publication. I have one additional comment, however. IN line 32 of the revised version, they state that PMW have floats up to 15 cm. This is a considerable understatement. The PMW found in the Atlantic can have floats up to about 30 cm long and 15 cm high (see http://www.americanoceans.org/species/portuguese-man-o-war). I have seen samples this size on the beach at Corpus Christi, Texas, and also from a research boat in the northeastern Gulf of Mexico. This large size difference from the bluebottles discussed in the paper may account for the difference in coefficients used by the authors compared with those used by Iosilevskii and Weihs.

**Author:**
We thank the editor for their comment. Since reviewer #2 specified that the size of the PMW float is usually less than 15 cm, we have rewritten the statement to take into account the whole range.

**Author's changes:**
The manuscript now reads (line 30-32) "In comparison, the PMW has floats of around 15 cm, reported up to 30 cm, and several hunting tentacles that can reach 30 m in mature colonies when fully extended (Munro et al. (2019))."

We have also corrected a small mistake in lines 362-363 of the latest submission. All calculations were thoroughly checked.